Isolation and characterization of eight polymorphic microsatellites for the spotted spiny lobster, Panulirus guttatus

Truelove Nathan 1 4 trueloven@si.edu
Behringer Donald C. 2
Butler IV Mark J. 3
Preziosi Richard F. 1
1 Faculty of Life Sciences, University of Manchester , Manchester , England
2 School of Forest Resources and Conservation & Emerging Pathogens Institute, University of Florida , Gainesville, FL , United States
3 Department of Biological Sciences, Old Dominion University , Norfolk, VA , United States
4 Current affiliation: Smithsonian Museum of Natural History, Smithsonian Marine Station , Fort Pierce, FL , United States
Toonen Robert
Electronic publication date: 2016 Jan 25
Publication date: 2016
Volume: 4
Electronic Location ID: e1467
Received 2014 Apr 27; Accepted 2015 Nov 12
Copyright: ©2016 Truelove et al.
Copyright year: 2016
Copyright holder: Truelove et al.
License: This is an open access article distributed under the terms of the Creative Commons Attribution License, which permits unrestricted use, distribution, reproduction and adaptation in any medium and for any purpose provided that it is properly attributed. For attribution, the original author(s), title, publication source (PeerJ) and either DOI or URL of the article must be cited.
License URL: https://creativecommons.org/licenses/by/4.0/

Keywords: Conservation, Population genetics, Neutral marker, Connectivity

Funding: University of Manchester NSF OCE0929086 NKT received funding from the University of Manchester, Sustainable Consumption Institute. This work was funded in part by NSF grant OCE0929086 to MJB and DCB. The funders had no role in study design, data collection and analysis, decision to publish, or preparation of the manuscript.

==============================
Microsatellite sequences were isolated from enriched genomic libraries of the spotted spiny lobster, Panulirus guttatus using 454 pyrosequencing. Twenty-nine previously developed polymerase chain reaction primer pairs of Panulirus argus microsatellite loci were also tested for cross-species amplification in Panulirus guttatus. In total, eight consistently amplifying, and polymorphic loci were characterized for 57 individuals collected in the Florida Keys and Bermuda. The number of alleles per locus ranged from 8 to 20 and observed heterozygosities ranged from 0.409 to 0.958. Significant deviations from Hardy–Weinberg equilibrium were found in one locus from Florida and three loci from Bermuda. Quality control testing indicated that all loci were easy to score, highly polymorphic and showed no evidence of linkage disequilibrium. Null alleles were detected in three loci with moderate frequencies ranging from (20% to 22%). These eight microsatellites provide novel molecular markers for future conservation genetics research of P. guttatus.

Introduction

The spotted spiny lobster Panulirus guttatus is a coral reef dwelling species that occurs from Bermuda to Suriname and throughout the Caribbean Sea (Sharp, Hunt & Lyons, 1997). P. guttatus is believed to have a long pelagic larval duration and occupies the same coral reef habitat throughout all of its benthic stages (Sharp, Hunt & Lyons, 1997). P. guttatus matures at a relatively small size compared to other species of spiny lobster (females 32 mm carapace length (CL), males 36–37 mm CL). Despite its small size, fishing pressure has begun to increase due to declining Panulirus argus fisheries in the Caribbean (Wynne & Coté, 2007; Fanning, Mahon & McConney, 2011). Fishery regulations for P. guttatus are either extremely limited (e.g., Bermuda and Martinique) or non-existent, and fisheries are emerging in the British West Indies and several other Caribbean nations to satisfy the demand for luxury seafood (Acosta & Robertson, 2003; Wynne & Coté, 2007). Management is hindered by a lack of basic life history, ecology, and population information—all of which would be facilitated by the development of species-specific genetic tools.

This study aims to enable future genetic studies on P. guttatus by characterizing new microsatellites for the species. Whilst microsatellites have already been developed for several spiny lobster species from the genus Panulirus (Decapoda: Palinuridae) (Ben-Horin et al., 2009; Kennington et al., 2010; Dao, Todd & Jerry, 2013; Liu, Yang & Liu, 2013), only microsatellites for P. argus (Diniz et al., 2004; Diniz et al., 2005; Tringali, Seyoum & Schmitt, 2008) were only tested for cross-species amplification since the success rate of amplification in more distantly related congeners (Ptacek et al., 2001) is generally low (Ben-Horin et al., 2009). These microsatellite primers will allow researchers to identify genetically unique subpopulations, determine levels of genetic diversity, and measure levels of genetic connectivity among subpopulations of P. guttatus.

Methods

The authors collected DNA samples from P. guttatus, completed DNA extractions, and tested validated microsatellite loci for polymorphism. Genoscreen, France (www.genoscreen.fr) tested P. guttatus DNA for quality and quantity, developed microsatellite libraries, performed 454 pyrosequencing, used bioinformatics software to identify potentially amplifiable microsatellite loci, and validated potentially amplifiable loci. Leg muscle tissue was collected from 24 individuals from Long Key Florida and 33 individuals from North Rock Bermuda. Total genomic DNA was isolated with the Wizard SV-96 Genomic DNA extraction kit (Promega, Madison, WI, USA). Genomic DNA from 12 individuals from Long Key Florida was used by GenoScreen for microsatellite development. The DNA quantity was assessed using the Picogreen assay (Invitrogen, Carlsbad, California, USA). To improve polymorphism detection the DNA from 12 individuals were pooled equimolarly. Microsatellite libraries were developed using 1 µg of pooled DNA and 454 GS FLX Titanium pyrosequencing of the enriched DNA (Malausa et al., 2011). Briefly, total DNA was enriched for microsatellite loci using 8 probes (AG, AC, AAC, AAG, AGG, ACG, ACAT and ATCT) and subsequently amplified. The PCR products were purified, quantified, and GS FLX libraries were developed following the manufacturer’s protocols (Roche Diagnostics, Risch-Rotkreuz, Switzerland) and sequenced on a GS FLX-PTP. The level of coverage used to develop microsatellite loci was 1/32 of a plate. This technique allowed the identification of 12,676 potential microsatellite primers. The bioinformatics program QDD was used (Meglécz et al., 2010) to identify sequences that were optimal for primer design and validated 737 pairs of primers. Tri-repeats and tetra-repeats were favored in order to minimize stutter bands and increase the probability of accurate allele scoring. The following selection parameters were used to design microsatellite primers: minimum melting temperature (Tm) of 60°C; optimum Tm of 71°C; maximum difference in Tm between primer pairs of 5°C; and primer length of 20–30 bp. Twenty-four validated sets of P. guttatus primers and 29 sets of previously designed microsatellite primers for P. argus (Diniz et al., 2004; Diniz et al., 2005; Tringali, Seyoum & Schmitt, 2008) were tested for amplification. Primer sets were discarded if they either failed to amplify or amplified three or more distinct fragments. The 13 microsatellites developed by Genoscreen and 2 microsatellites previously developed (Tringali, Seyoum & Schmitt, 2008) for P. argus were tested for polymorphism in P. guttatus using the forward labeled fluorescent primers 6-FAM, HEX, NED, and PET. Twelve of the thirteen microsatellites identified by Genoscreen were partitioned into 3 multiplexes consisting of 4 primer pairs. The forth multiplex consisted of a Genoscreen primer pair and the two primer pairs developed for P. argus (Tringali, Seyoum & Schmitt, 2008). A unique fluorescent label was attached to the forward primers of each multiplex. Annealing temperatures for all primer pairs were calculated with Multiplex Manager (Holleley & Geerts, 2009) using 200 nanomolar primer concentration and 10°C below the primer melting temperature Tm.

Each multiplex PCR was performed with a Veriti thermal cycler (Applied Biosystems, Foster City, CA, USA). Our protocol followed the manufacturer’s recommendations (Qiagen Microsatellite Multiplex PCR Kit; Qiagen, Hilden, Germany), however, we initially compared reaction volumes of 25 µl, 10 µl and 5 µl for each multiplex in 24 individuals. Results were identical for each reaction volume, therefore to reduce costs the total volume of the PCR reaction was scaled down from 25 µl to 5 µl whilst keeping the concentrations of all PCR reagents the same. The PCR reaction mix consisted of 0.5 µl of the 10X primer mix (1 µM primer + 1 µM fluorescent primer), 2.5 µl of Type-it Multiplex PCR Master Mix (Qiagen, Hilden, Germany), 1 µl of molecular grade water and 1 µl of (10–20 ng/µl) genomic DNA. The PCR conditions consisted of an initial denaturation at 95°C for 5 min, followed by 26 cycles at 95°C for 30 s, 59–65°C for 120 s (the lowest primer annealing temperature was chosen for each multiplex; Multiplex 1 = 58°C, Multiplex 2 = 62°C, Multiplex 3 = 60°C, Multiplex 4 = 65°C), and 72°C for 30 s. This was followed by final extension at 60 °C for 30 min. To facilitate the fragment analysis, PCR products were diluted 1:1 with 5 µl MQ water. From the diluted product, 0.5 µl was mixed with 9.5 µl of a mix consisting of Hi-Di Formamide® (Applied Biosystems, Foster City, CA, USA) and GeneScan—500 LIZ Size Standard (37:1) in a 96 well PCR plate. Fragment analysis was performed on an ABI 3730xl automatic DNA sequencer (Applied Biosystems, USA) at the University of Manchester DNA Sequencing Facility. Microsatellite alleles were scored using the GeneMapper® v3.7 software package (Applied Biosystems, Foster City, CA, USA). Binning of microsatellite alleles and error checking were preformed using the R package MsatAllele version 1.02 (Alberto, 2009) and R statistical software v2.15.1 (Ihaka & Gentleman, 1996). The entire data set was checked for variability and departures from Hardy–Weinberg equilibrium (HWE) and the fixation index (FIS) was calculated using the software package Genodive v2.0b23 (Meirmans & Van Tienderen, 2004; Meirmans, 2012). The Benjamini Hochberg method (i.e., the false discovery rate) was used to correct for multiple comparisons of HWE (Benjamini & Hochberg, 1995). Linkage disequilibrium between loci was tested using Genepop on the Web v4.2 (Raymond & Rousset, 1995; Rousset, 2008). Markov chain parameters for Genepop were set to the following: dememorization number 10K, number of batches 1K, and number of iterations per batch 10K. Null allele frequencies and scoring errors caused by stutter peaks or large allele dropout was calculated with MICROCHECKER (Van Oosterhout et al., 2004).

Table 1 Characterization of eight microsatellite loci for Panulirus guttatus with GenBank, GenBank Accession Number; TA, annealing temperature; Na, number of alleles; Ho, observed heterozygosity; He, expected heterozygosity; Fis, fixation index; P, P-value for deviation from Hardy-Weinberg equilibrium and FNA, null allele frequency. Fluorescent labels on forward primers and significant values after the false discovery rate correction for multiple comparisons (Benjamini & Hochberg, 1995) are in bold.

Locus	Primer sequence (5′–3′)	Multiplex	GenBank	Repeat motif	Range (bp)	TA	Florida (N = 24)					Bermuda (N = 33)		
							Na	Ho	He	Fis	P	FNA	Na	Ho	He	Fis	P	FNA	
Pgut-3	F: GCTGGAGAGGGAGGAACTGT-6FAM	1	KC800822	(GAG)12	95–131	66.7	16	0.696	0.843	0.175	0.088	–	11	0.667	0.892	0.253	<0.001	0.141	
	R: CCCTTCCTCATCTTTCTTCTCC					63.3													
Pgut-6	F: CCCATTCATTTTCGTCATCA-PET	1	KC800823	(ATC)12	140–165	58.3	11	0.75	0.872	0.139	0.208	–	10	0.710	0.855	0.170	0.589	–	
	R: CCTTGATTTCAAATTGCTGC					59.4													
Pgut-9	F: GTGTGGTTGTTGACGTTGCT-VIC	2	KC800824	(TGT)17	78–119	64.6	8	0.667	0.834	0.201	0.06	–	11	0.938	0.758	0.236	0.727	–	
	R: GACTCGAAGACGCAGACGTA					63.5													
Pgut-15	F: CACCAGTTGTGAAAATACTTTTGCT-PET	2	KC800825	(GATA)6	133–178	63.3	13	0.958	0.836	0.146	0.365	–	12	0.844	0.856	0.014	0.142	–	
	R: GTCCTAGAAAAGATAAAAGCTTAGGGA					62.4													
Pgut-21	F: TGCCCTTGGCAAAATCTCTA-VIC	3	KC800826	(TCTA)8	167–224	60.6	9	0.875	0.829	0.055	0.711	–	12	0.697	0.823	0.153	<0.001	0.204	
	R: GCGAACTGAACGCTTCCTAA					62.7													
Pgut-22	F: CCTTGCATCCCAGACGTGTA-6FAM	3	KC800827	(TGTA)10	74–115	64.3	11	0.5	0.839	0.404	0.129	–	9	0.5	0.837	0.403	<0.001	0.217	
	R: ACGCGGACACATACTCTCCT					65.7													
Pgut-23	F: AAGGAAATAGCCTCGCCAAT-NED	3	KC800828	(AGAT)11	133–171	62.5	10	0.409	0.844	0.515	<0.001	0.22	8	0.613	0.765	0.199	0.366	–	
	R: AATGGGTACCTGGCTCAAGA					62.9													
Par-Fwc05	F: AGAGAGACGCTGCTGTTCTTC-6FAM	4	EF620542	(CA)18C(CA)10	131–179	65.1	8	0.583	0.753	0.226	0.041	–	20	0.781	0.919	0.150	0.041	–	
	R: AAAGGGCATCCTCGGTAGAGTC					66.7													

Results

Six out of 13 microsatellites developed by Genoscreen were found to be either monomorphic or too difficult to score and were removed from the analysis. Twenty-seven out of 29 P. argus microsatellites failed to produce PCR products. One out of the two P. argus microsatellites that did produce a PCR product was too difficult to score and was removed from the analysis. Table 1 summarizes the characteristics of the eight primer pairs of polymorphic and easy to score microsatellite loci developed for the spotted spiny lobster P. guttatus. No evidence of linkage disequilibrium was found among any of these loci. We were unable to test for Mendelian inheritance since crossbreeding of P. guttatus has yet to be achieved under laboratory conditions.

Samples from Long Key Florida (N = 24) and Bermuda (N = 33) were genotyped using the eight developed primers. The number of alleles ranged from 8 to 20 per locus. Significant deviations from Hardy–Weinberg equilibrium were found in one locus from Florida and three loci from Bermuda (Table 1). These deficiencies could be due to null alleles or the Wahlund effect (Johnson & Black, 1984). The latter is possible considering the potential for extensive geneflow in this species. However, null alleles are a common characteristic of the microsatellites of many marine invertebrates, so could also be responsible for the deviations from HWE (Dailianis et al., 2011). Indeed null alleles were detected by MICROCHECKER in the four loci that deviated from HWE (Par-Fwc05, PG3, PG21, PG22). Null allele frequencies in these loci ranged from 20% to 22% (Table 1). Although null alleles have been found to inflate levels of population structure, they do not create population structure where it does not already exist (Chapuis & Estoup, 2007; Carlsson, 2008).

Discussion

Population genetics studies have yet to be conducted on this coral reef lobster species that is facing increasing fishing pressure. Even though 4 microsatellite primers show evidence of null alleles, the moderate null allele frequencies and number of alleles suggests these eight primers are useful for conducting future genetic studies of P. guttatus. This research can be used to help develop conservation and fishery management plans for this understudied species.

Supplemental Information

Dataset S1 Revised dataset

Microsatellite data for Panulirus guttatus in the Genepop format.

Click here for additional data file.

We thank Dr. Tammy Trott from the Bermuda Fisheries Department for providing samples for this study and Josh Anderson, Jason Spadero, and Mike Dixon for helping to collect samples in the Florida Keys. We are grateful to Antoine Destombes at Genoscreen for his help with this project.

Additional Information and Declarations

Competing Interests

Author Contributions

Field Study Permissions

DNA Deposition

Data Availability

The authors declare there are no competing interests. Nathan Truelove is currently a Postdoctoral Fellow at the Smithsonian Marine Station in Fort Pierce.

Nathan Truelove conceived and designed the experiments, performed the experiments, analyzed the data, contributed reagents/materials/analysis tools, wrote the paper, prepared figures and/or tables, reviewed drafts of the paper.

Donald C. Behringer and Mark J. Butler IV conceived and designed the experiments, contributed reagents/materials/analysis tools, reviewed drafts of the paper, provided Assistance Collecting Samples.

Richard F. Preziosi conceived and designed the experiments, contributed reagents/materials/analysis tools, reviewed drafts of the paper, provided Funding for Lab Reagents.

The following information was supplied relating to field study approvals (i.e., approving body and any reference numbers):

Special Activity License from the Florida Fish and Wildlife Conservation Commission, Division of Marine Fisheries Management, License Number: SAL-10-0582-SR.

The following information was supplied regarding the deposition of DNA sequences:

Genbank accession numbers: KC800822, KC800823, KC800824, KC800825, KC800826, KC800827, KC800828.

The following information was supplied regarding data availability:

The data set used in this study is available as Dataset S1.

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
