# Peer review of "Isolation and characterization of eight polymorphic microsatellites for the spotted spiny lobster, Panulirus guttatus"

_PeerJ, doi:10.7717/peerj.1467_

## Round 0.1 · original submission · Major Revisions

· Academic Editor

Major Revisions

I have now gotten responses from 3 referees on your manuscript, but one of them is so brief that it is simply uninformative. The most careful referee has extensive comments and recommendations for improvement of the manuscript. I find myself in agreement with the first referee on most of the points that they raise, particularly the issue of HWE and null alleles. It seems like a simple addition to include Microchecker and/or FreeNA to examine the cause of these deviations or possible impact on the analyses as suggested by the referee. Additionally, Selkoe & Toonen (2006) provide a detailed review on quality control testing in the Microsatellites for Ecologists manuscript if you need further detail from the guidance provided in the review. Because I agree with the referee that the manuscript would benefit from some additional analyses, I have listed this as a major revision. However, I believe that the suggestions of the referee are relatively straight forward and you should be able to revise the manuscript in such a way that it would become acceptable if you follow their valuable advice.

Reviewer 1 ·

Basic reporting

I thought there were a number of things missing that are traditionally found in microsatellite primer notes. My suggestions follow:

Abstract
The abstract should be modified to more accurately reflect the study:

1. In the abstract, it says 122 individuals were characterized using these microsatellites, but in the rest of the paper, it seems that only 77 were – please clarify

2. It would be good to mention that the microsatellites were developed for P. guttatus using 454 pyrosequencing

3. Significant deviations from HW equilibrium could be caused by null alleles, but this is not the only reason that deviations occur (e.g. Wahlund effect, inbreeding, selection), so it's not correct to say that deviations from HWE suggest null alleles. I would rephrase this to reflect that null alleles are one reason for seeing deviation from HWI, or elaborate on further tests you did that suggest the deviations are caused by null alleles.

4. It’s also confusing that you mention in one sentence that there are significant deviations from HWE due to null alleles, and in the next say that there’s no significant deviation from genotypic equilibrium and that null alleles are at low to moderate frequency.

5. Finally, the last sentence states that these microsatellites should provide sufficient statistical power to detect fine scale genetic structure, but there’s no power analysis done in this paper for these microsatellites that I can see.

Introduction
It struck me as odd that P. argus is mentioned in the introduction almost as much as P. guttatus. I certainly know the importance of P. argus as a fishery species, and the rationale for studying P. guttatus as fishing pressure increases on this species makes sense (lines 22-26). However, it seems there should be enough justification for studying P. guttatus on its own that the rest of the introduction (lines 1-21) could focus mostly on this species (i.e. no need for discussing nursery habitat, movement patterns, etc. of P. argus). One suggestion might be to shorten this section and not separate the Introduction and Methods sections, as there’s no Results/Discussion sections.

Lines 30-35: Microsatellites have also been developed for Panulirus cygnus (Kennington et al. 2010, Con Gen Res), P. interruptus (Ben-Horin et al. 2009, Con Gen Res), P. ornatus (Dao et al. 2013 Con Gen Res; Liu et al. 2013, Journal of Genetics), and P. stimpsoni (Liu et al. 2010, Hereditas). While I would not expect the authors to test all of these, some statement of rationale as to why P. argus microsatellites were tested for cross-species amplification, but not other species should be made here (i.e. phylogenetic relatedness (Ptacek et al. 2001, Marine and Freshwater Research), and the generally low rate of successful amplification in more distantly related congeners – which has a variety of citations that could be used).

Methods
I think the methods could use some clarification overall – especially since this is essentially a methods paper. For example, which parts did the authors complete, and which were carried out by GenoScreen? This could be easily rectified with a brief sentence. If 25 individuals were sent to GenoScreen for development (line 42), and then 12 were pooled equimolarly to improve polymorphism (line 44-45), what was done with the other 13? Or were these 12 in addition to the 25 initially sent? If only 12 were pooled, why weren’t the other 13 genotyped for allele counts, heterozygosity, etc.?

What level of coverage for 454 sequencing was used to develop the loci – 1/16th of a plate? A full plate?

Which program was used for primer design?

What were the fluorescent primers used?

No results are reported for linkage disequilibrium. Were any of these loci linked?

Was Mendelian inheritance tested for each of these markers? If not, why not?

Line 40-41: Are these 25 individuals in addition to the 49 from which DNA was isolated in line 37? If so, then this would make the sum of 122 mentioned in the abstract. However, then it wouldn’t make sense that only 24 individuals from Long Key were genotyped to examine the number of alleles, heterozygosity, etc.

Line 52: What was considered “sequences that were optimal for primer design”? What were the selection parameters chosen?

Line 84-85: If “Introduction” and “Methods” sections are kept, it seems that a “Results” division could go prior to line 85.

Line 92-93: Check your sample #s. In line 38, it says DNA was isolated from 48 individuals from Bermuda, here it says 50 individuals were genotyped. Also clarify the Long Key numbers as noted above.

Figure 1 is interesting, but probably unnecessary. If kept, it should have a more detailed explanation for the darker bars versus single bars – I’m assuming this is variation around the sizes for each allele and the main breaks are where the bin sizes were set?

In Table 1, please write in the number of repeats in the repeat motif that were sequenced in the initial clone (i.e. (gag)12). Also list the annealing temperature used for each marker. In my version, a couple of the headings move to a second line – make sure the table columns are wide enough so this does not happen in the final version.

Experimental design

Overall, the experimental/microsatellite design seemed fairly standard using 454-pyrosequencing. Beyond some of the basic reporting issues I have listed above, I did have a couple of questions about some of the marker evaluation. Specifically, I think the authors could do a more complete job of evaluating what's driving the deviation from HWE in the majority of their markers. While deviations in HWE are fairly common for marine invertebrates, it's not ideal to have 7/8 markers out of HWE. I think with a bit more effort, the authors could at least have a better understanding of what's driving these patterns in this system. Detailed comments are below on various aspects.

Line 59-63: 5ul reactions seems very small, even for microsatellites. Did you test larger reaction volumes to see if this had any influence on the null allele amplification? Were these multiplexed microsatellite runs for the polymorphism tests, or each microsatellite marker was amplified individually, but pooled for genotyping? Or genotyped individually as well?

Line 95: How was it determined that all of the deviations from HWE were from heterozygote deficiencies?

Line 96: This can be tested for more explicitly – For example Micro-checker (vanOosterhout et al. 2004) can be used to determine whether there are scoring errors due to the way stutter-peaks are scored or to large allele drop-out, and provide an estimate of null allele frequencies. With multi-locus genotypes, as you would have here, it can discriminate between HWE deviations caused by null alleles vs. Wahlund effects vs. inbreeding. A few loci have high FIS values (i.e. Pgut-3, P-gut 22). Are these also the loci that have high null allele frequencies (22 – yes). A strong correlation between FIS and null allele frequency would make a stronger case that this is the cause of the HWE deviations.

Validity of the findings

The findings of 8 microsatellites that reliably amplify in P. guttatus appears valid. I would remove the line from the abstract though that states that these microsatellites should provide sufficient statistical power to detect fine scale genetic structure, unless a power analysis is conducted to test this.

Additional comments

Minor Comments:
Line 49 and 50: GsFLX should be GS-FLX as in line 46? Or all GsFLX?
Line 56: space after “Tringali et al 2008)”
Line 56-57: Change the wording for “Primer sets were discarded if they failed to amplify or lead to >2 fragments.” As worded, this could be taken as having 2 different meanings.
Line 58: I would move the Tringali citation to after “previously developed”.

Reviewer 2 ·

Basic reporting

The manuscript is well written

Experimental design

The experiments were designed appropriately and the methods were described substantially.

Validity of the findings

The findings were acceptable

Additional comments

There are number of typo-errors.

·

Basic reporting

1. Corresponding author is not mentioned in the text;
2. In-text citations do not comply PeerJ format policies;
3. In-text citations out of chronological order (e.g. line 29);
4. References in the text do not comply PeerJ format policies;
5. Not all units are in the SI (e.g. ml...mL);
6. Typing mistakes (e.g. line 39);
7. Figure 1 could be removed (at editor's discretion).

Experimental design

The article meets PeerJ standards.

Validity of the findings

Results indicate that the eight microsatellite loci would be useful as molecular markers in genetic studies on P. guttatus. Results are in accordance with objective and methodology. Findings meet PeerJ standards. No additional comments.

Additional comments

This short article is of general interest for conservation geneticists or fishery managers of this spiny lobster species, even though few microsatellite loci (8) are suggested to be useful as molecular markers. Some corrections should be made in the text, but mostly format-related.

---

## Round 0.2 · accepted · Accept

· Academic Editor

Accept

Thank you for your revisions of the manuscript in response to the referee suggestions. I have read through the manuscript and am satisfied with your response and revisions, such that I am happy to move the manuscript forward. I note that there remain some typos that you will want to correct in the proofs (e.g., "...only microsatellites for P. argus where only tested for cross-species amplification...") but the manuscript is sufficiently improved to move it forward into production.